# Spirituality, Community Belonging, and Mental Health Outcomes of Indigenous Peoples during the COVID-19 Pandemic

**DOI:** 10.3390/ijerph19042472

**Published:** 2022-02-21

**Authors:** Chantal Burnett, Eva Purkey, Colleen M. Davison, Autumn Watson, Jennifer Kehoe, Sheldon Traviss, Dionne Nolan, Imaan Bayoumi

**Affiliations:** 1Faculty of Health Sciences, School of Medicine, Queen’s University, Kingston, ON K7L 3N6, Canada; c.burnett@queensu.ca (C.B.); eva.purkey@queensu.ca (E.P.); davisonc@queensu.ca (C.M.D.); dwweast@idhc.life (A.W.); 2Indigenous Health Council, Kingston, ON K7K 2V4, Canada; jennybug_1977@hotmail.com (J.K.); sheldonktraviss@gmail.com (S.T.); dionne.nolan@hotmail.com (D.N.); 3Indigenous Diabetes Health Circle, Thorold, ON L2V 4Y6, Canada

**Keywords:** Indigenous Peoples, mental health, spirituality, social support, culturally competent care

## Abstract

We aimed to assess the association between community belonging, spirituality, and mental health outcomes among Indigenous Peoples during the COVID-19 pandemic. This cross-sectional observational study used online survey distribution and targeted outreach to the local Indigenous community to collect a convenience sample between 23 April 2020 and 20 November 2020. The surveys included demographic information, self-reported symptoms of depression (PHQ-2) and anxiety (GAD-2), and measures of the sense of community belonging and the importance of spirituality. Multivariate logistic regression was used to model the association between the sense of community belonging and spirituality, and symptoms of anxiety and depression. Of the 263 self-identified Indigenous people who participated, 246 participants had complete outcome data, including 99 (40%) who reported symptoms of depression and 110 (45%) who reported symptoms of anxiety. Compared to Indigenous participants with a strong sense of community belonging, those with weak community belonging had 2.42 (95% CI: 1.12–5.24)-times greater odds of reporting symptoms of anxiety, and 4.40 (95% CI: 1.95–9.89)-times greater odds of reporting symptoms of depression. While spirituality was not associated with anxiety or depression in the adjusted models, 76% of Indigenous participants agreed that spirituality was important to them pre-pandemic, and 56% agreed that it had become more important since the pandemic began. Community belonging was associated with positive mental health outcomes. Indigenous-led cultural programs that foster community belonging may promote the mental health of Indigenous Peoples.

## 1. Introduction

For decades, adverse mental and physical health impacts of colonialism for Indigenous Peoples in Canada have been reported [1,2]. Colonial policies and systems had goals to assimilate or eradicate Indigenous Peoples, and were broad in scope [3,4]. Canada established a reserve system, in which band members lacked voting rights and rights to access higher education, own businesses or serve in the military. The government mandated the removal of children from their families and communities to residential schools for the expressed purpose of assimilation. In residential schools, children were forbidden to use their traditional cultural practices or speak Indigenous languages, and suffered widespread physical, emotional, sexual and spiritual abuse, as well as very high mortality rates [3,5]. More recently, the Canadian government has been censured for racially discriminating against First Nations children due to its underfunding of social services and child welfare services on reserves. After 20 non-compliance orders against them, the Canadian government negotiated a settlement to compensate affected individuals, and stated a commitment to major reforms [6]. These historic traumas and ongoing adversities related to social determinants of health, adverse childhood experiences, and discrimination have led to mental health disparities between Indigenous Peoples and non-Indigenous people [7].

Disparities in mental health are compounded by the ongoing COVID-19 pandemic, which has resulted in large-scale social disruption, and has left many to struggle with changes to routines and feelings of uncertainty. Thakur and Jain (2020) refer to it as a “global psychological pandemic”, as COVID-19 and its social disruptions have been associated with symptoms of anxiety, stress, and depression [8].

In a recent survey by Statistics Canada, 60% of Indigenous respondents indicated that their mental health had worsened since physical distancing began. The survey also found that higher proportions of Indigenous respondents reported fair or poor mental health compared to non-Indigenous respondents (38% vs. 23%). Additionally, 41% of Indigenous respondents reported symptoms consistent with moderate or severe anxiety (compared to 25% of non-Indigenous respondents) [9].

Cultural identity, connectedness, and spirituality have been shown to be associated with the positive mental health outcomes of Indigenous Peoples [4,10]. There is also evidence that an individual’s sense of community belonging, which describes the degree to which they feel connected to their community and place within it, is associated with positive mental health outcomes, independent of geography and socioeconomic status [11,12,13]. The objective of this study was to assess the association between a sense of community belonging and spirituality during the COVID-19 pandemic and mental health outcomes among urban Indigenous Peoples.

## 2. Materials and Methods

### 2.1. Study Design and Participants

This was a cross-sectional observational study that used survey data from the Cost of COVID study, which aimed to understand the social and emotional impacts of COVID-19.

An online anonymous survey was used, which included demographic characteristics and various measures of social and emotional well-being. The survey was distributed widely through professional and social networks, as well as social media, by investigators in Kingston. To help mitigate selection bias, we conducted targeted outreach to local services for homeless individuals and emergency food programs, and Indigenous community members. The data were collected between 23 April 2020 and 30 November 2020.

The recruitment of Indigenous participants was supported by an Indigenous Research Associate team member, who disseminated the invitations to participate both electronically and in-person, when in-person recruitment was possible. The recruitment was also guided and supported by a sub-committee of the local Indigenous Health Council. The involvement of trusted community members was instrumental to the participant recruitment.

### 2.2. Measurements

The Cost of COVID survey included items asking for demographic information such as age, gender, education, and household annual income. In order to measure the sense of community belonging, the participants were asked to use a Likert scale (ranging from Very Strong, Somewhat Strong, Somewhat Weak, Very Weak) to answer the following question: “How would you describe your sense of belonging to your local community?” The participant responses were categorized as ‘Strong’ (including Very Strong and Somewhat Strong) or ‘Weak’ (including Very Weak and Somewhat Weak).

In order to measure the importance of spirituality before and during the COVID-19 pandemic, the participants reported the extent of their agreement with the statements, “Spirituality was important to me before the pandemic” and “Spirituality has become more important to me since the pandemic began”. The response options included Strongly Agree, Agree, Neither Disagree nor Agree, Disagree, and Strongly Disagree. The responses were categorized as ‘Agree’ (including Agree and Strongly Agree) and ‘Disagree’ (including Strongly Disagree, Disagree, and Neither Agree nor Disagree).

In order to assess symptoms of depression and anxiety, the Patient Health Questionnaire-2 (PHQ-2) and the Generalized Anxiety Disorder 2-item (GAD-2) were included in the survey. The PHQ-2 and the GAD-2 are validated symptom-based psychiatric rating scales and validated screening tools for depression and anxiety in primary care settings [14,15].

In the PHQ-2, the respondents are asked to estimate the frequency of two symptoms (depression and anhedonia) over the past two weeks, with four response options ranging from “not at all” to “nearly every day”, and a final score ranging from 0 to 6. The items are derived from questions designed to establish DSM-IV psychiatric diagnoses [15]. A PHQ-2 score cut-point of 3 has been shown to have a sensitivity of 83% and a specificity of 90% for the detection of major depressive disorder (MDD) [16]. Similarly, Löwe et al. (2005) found that a PHQ-2 score cut-point of ≥3 has the best trade-off between sensitivity and specificity for both MDD and any depressive disorder [17]. Together, these studies provide strong evidence for the validity of the PHQ-2 as a depression screening tool.

The GAD-2 also asks respondents to estimate the frequency of two symptoms (nervousness and the ability to control worrying) over the past two weeks, with the same response options and final score range as the PHQ-2. A blinded study comparing the GAD-2 scale in primary care settings with psychiatric diagnoses based on the DSM-IV criteria found that the GAD-2 had a high sensitivity (86%) and specificity (83%) for GAD, and a high specificity for social anxiety disorder (81%) and any anxiety disorder (88%) (based on a GAD-2 score cut-point of 3) [16].

### 2.3. Statistical Analysis

The analysis was comprised of descriptive statistics and multivariate logistic regression models to assess the association between the sense of community belonging and mental health outcomes (symptoms of anxiety and depression), and the association between the importance of spirituality and the same mental health outcomes.

In order to assess for the influence of potential confounders, we evaluated the association after adjusting the models for age, gender, highest level of education completed, and household annual income. These variables were selected based on literature demonstrating their influence on mental health status [18,19]. All of the quantitative analyses were completed in SAS^®^ Studio.

### 2.4. Ethics

This study was approved by the Queen’s Health Sciences Research Ethics Board. In order to ensure effective and respectful data collection and analysis, the First Nations principles of OCAP^®^ (Ownership, Control, Access, and Possession) were applied throughout the entire research process. The OCAP^®^ principles represent a set of standards that establish the ways in which First Nations data should be collected, protected, used, and shared (i.e., the standard for the ways in which to conduct research with First Nations people). These principles assert that First Nations people have control over data collection processes in their communities, and that they own and control the ways in which this information can be used [20]. To abide by these principles, our team consulted regularly with the sub-committee of the local Indigenous Health Council (DN, JK, ST) in order to operationalize OCAP^®^. The intention of these meetings was to discuss and review the research findings, and to support the interpretation of the results and the knowledge translation.

## 3. Results

In total, 263 self-identified Indigenous people participated in the Cost of COVID study (Table 1). Most of the participants were younger than 50 years of age (74%), identified as female (88%), and had some post-secondary education (68%). The participants’ household incomes were distributed across the income categories. Of note, 12% of the respondents were precariously housed (staying with friends or family members, emergency shelters, or unsheltered). This may have reflected the sampling strategy, which included targeted recruitment from services for homeless individuals and emergency food programs.

There were 246 Indigenous participants who responded to the GAD-2 and PHQ-2 survey questions—110 (45%) reported symptoms of anxiety, and 99 (40%) reported symptoms of depression. With regards to the sense of community belonging, 100 of the 171 (58%) Indigenous participants we received responses from described it as “somewhat strong” or “very strong”. Of the 174 Indigenous participants who responded to questions about spirituality, 132 (76%) reported that spirituality was important to them before the pandemic, and 97 (56%) reported that it became more important to them since the pandemic began. Of the 42 Indigenous participants reporting that spirituality was not important pre-pandemic, 6 (14%) indicated that it became more important during the pandemic.

The results from the adjusted logistic regression models are reported in Table 2 and Table 3. Compared to participants with a strong sense of community belonging, those with a weak sense of community belonging had 2.42 (95% CI: 1.12–5.24)-times greater odds of reporting symptoms of anxiety, and 4.40 (95% CI: 1.95–9.89)-times greater odds of reporting symptoms of depression. The importance of spirituality during the pandemic was not significantly associated with symptoms of anxiety (aOR 0.7, 95% CI: 0.4–1.5) or depression (aOR 1.0, 95% CI: 0.5–2.0). In order to help contextualize the results, we assessed changes in access to Indigenous cultural services, and found that fewer participants accessed such services after the pandemic began (Table 4).

## 4. Discussion

There was a high prevalence of self-reported symptoms of depression (40%) and anxiety (45%) among the Indigenous participants of the Cost of COVID study. The participants with a weak sense of community belonging had more than two-fold higher odds of experiencing poor mental health. Having access to Indigenous-led, culturally specific programs that foster community belonging may be one avenue to promote mental health among urban Indigenous Peoples. The findings also highlight the growing importance of spirituality among a large proportion of participants within the context of a pandemic.

Our results align with literature examining the relationship between the sense of community belonging and mental health (both within and outside the context of a pandemic). Using data from the 2005 Canadian Community Health Survey (CCHS), Shields (2008) found that 64% of Canadians reported a strong sense of community belonging, and that it was strongly correlated with self-perceived mental health, after controlling for socioeconomic status, health behaviours, and the presence of health conditions [21]. This correlation is consistent across life stages and between regions across Canada [11,12].

The benefits of community belonging for mental health have also been observed specifically among other publications. For example, The McCreary Centre Society’s Adolescent Health Survey, which included 1710 Indigenous youth from British Columbia, found that youth who maintain positive family and school connections experience less emotional distress and have lower rates of risk-taking and suicidal behaviour [22]. In addition to social connectedness, there are also benefits of cultural connectedness (i.e., the engagement with Indigenous culture). Snowshoe et al. (2017) explored the association between cultural connectedness—which consists of identity, traditions, and spirituality—and the mental health outcomes of First Nations youth in Saskatchewan and Southwestern Ontario. The investigators found a strong association between cultural connectedness (as measured by the Cultural Connectedness Scale) and indicators of mental health, suggesting that cultural connectedness is an important contributor to Indigenous youths’ overall mental health and well-being [23].

More broadly, the association between social connectedness and mental health outcomes has also held true in the context of the COVID-19 pandemic. In their investigation of social connectedness and mental health during pandemic lockdown measures in Austria, Nitschke and colleagues (2021) found that a lack of social connectedness was correlated with perceived levels of stress, worry, and fatigue. The authors speculate that social connections provide a buffer against negative physical and mental health outcomes, and can promote resilience during times of uncertainty and distress [24].

These findings are not surprising, as humans are fundamentally a social species—our nature is to interact and form relationships, and loneliness or the perceived absence of social connection can impair executive functioning, sleep, and physical well-being [13]. Perceived loneliness is a risk factor for a host of mental and physical health problems, including depression, even after accounting for social support, demographic variables, dispositional negativity, and stress [13,22]. For Indigenous Peoples, community connections are integral with spirit. The Indigenous worldview says that everything has spirit (e.g., going to a sacred fire or the grandmother moon ceremony). Social connections are not just about connecting with other people but also about connecting with everything that surrounds us, as well as what is inside of us [25].

While the results highlight the importance of spirituality among a large proportion of Indigenous participants, the association between the importance of spirituality and mental health outcomes was not statistically significant, which contrasts with other published literature [10,26]. Our study may have been underpowered to detect a relationship that was in fact there. It is possible that our measure of spirituality lacked sufficient depth and discriminatory power. A more detailed measure of spirituality and spiritual practices may have better detected relationships between spirituality and mental health. In addition, the cohort may not have been representative of the population of Indigenous Peoples, which may have influenced this negative finding. The results may also be explained by the fact that, during pandemic lockdown measures, people’s ability to gather for spiritual ceremonies and gatherings was limited, and thus the association between spirituality and mental health in the specific context of the COVID pandemic, was not there. Indeed, the results indicate that since the COVID-19 pandemic began, fewer Indigenous participants accessed Indigenous cultural services (compared to before the pandemic) (Table 4).

Nonetheless, it remains essential to support the recommendations found in the report published by the Truth and Reconciliation Commission of Canada, Honouring the Truth, Reconciling for the Future (TRCC) [26]. The report highlights the importance of traditional practices—such as sweat lodges, cedar baths, smudging, the lighting of the Qulliq (a traditional oil lamp used by the Inuit), and other spiritual ceremonies—to Indigenous healing. It asserts that overcoming the health legacy of residential schools and closing the gap in health outcomes between Indigenous Peoples and non-Indigenous people requires the long-term empowerment of Indigenous communities to heal themselves. It recommends “best practices” for Indigenous wellbeing which involve the integration of mainstream healthcare and traditional practices, all under Indigenous leadership and control [26].

Our results indicate relatively commonly reported symptoms of anxiety (45%) and depression (40%) among Indigenous participants. Central to the provision of equitable mental healthcare for Indigenous Peoples is cultural safety—the act of providing services in a manner in which the relationship between the service provider and patient is built on a foundation of trust and respect, while considering power imbalances, institutionalized discrimination, and the impacts of colonialism on the patient’s health and well-being [27]. It is important to note that medicine and public health have been, and continue to be, a tool of colonization [3]. This has meant that mental health services have not always been culturally safe, thereby limiting Indigenous Peoples’ access to effective mental health support [28]. Moreover, access to safe spaces for individuals to conduct ceremonies and receive programming directed to spirit is limited within mainstream facilities. There is often a lack of understanding with regard to Indigenous needs and connection with the Indigenous community among non-Indigenous service providers.

For this reason, Calls to Action #21 and #22 of the TRCC report underscored the urgent need for self-determination in the use and access to traditional knowledge, therapies, and healing practices (Box 1) [26]. Indigenous-led approaches to mental healthcare are urgently needed because they are more likely to address inequities arising from the complex traumas faced by Indigenous communities today [10,28]. For example, in an analysis of Indigenous-led healthcare partnerships in Canada, Allen et al. (2020) found that Indigenous-led approaches were better suited to “revive, support and strengthen the worldviews and lifestyles underlying Indigenous conceptions of health and wellness, particularly in terms of nurturing the spirit” [28].

Box 1TRCC Calls to Action #21 and #22.
***21.** We call upon the federal government to provide sustainable funding for existing and new Aboriginal healing centres to address the physical, mental, emotional, and spiritual harms caused by residential schools, and to ensure that the funding of healing centres in Nunavut and the Northwest Territories is a priority.*

***22.** We call upon those who can effect change within the Canadian health-care system to recognize the value of Aboriginal healing practices and use them in the treatment of Aboriginal patients in collaboration with Aboriginal healers and Elders where requested by Aboriginal patients.*


Despite the TRCC Calls to Action and the urgent need for Indigenous-led health initiatives, there is a paucity of culturally safe mental health services in Canada. This discrepancy was echoed in surveys which found that while the majority of interviewed physicians in Ontario and British Columbia welcomed opportunities to learn about Indigenous approaches to healing, there is still a need for better access to culturally competent and safe healthcare services [29].

As is consistent with the OCAP^®^ principles, the data and summarized results of this study are being shared with Indigenous community members and service providers. This is occurring through the Indigenous Research Associate, members of the sub-committee of the Indigenous Health Council, a community knowledge dissemination circle, and the production of a series of infographics and oral presentations. The findings (the association between a strong sense of community belonging and improved mental health outcomes) will inform efforts by Indigenous community members, volunteers, and frontline workers to advocate for increased funding for the development and delivery of Indigenous-led programs and services that honour Indigenous knowledge, language, and culture, as well as improved access to culturally safe spaces within mainstream institutions. Community-based solutions to the fostering of community belonging may also be important to support improved mental health. Future research evaluating effective approaches to the strengthening of community belonging would help inform community development and service delivery for improved mental health.

The strengths of the present study include the application of a strength-based approach (rather than a deficit-based approach) and the use of validated mental health screening tools.

There are also some limitations for the present study. The online survey was shared through professional and social networks, as well as social media. This method of convenience sampling has many advantages, including the ability to distribute surveys widely in a timely manner, with minimal COVID-19 risk to the participants. To help mitigate selection bias, we purposively sampled in-person at several community sites in order to better capture marginalized populations in the sample. However, the urban-dwelling Indigenous people in the area come from many different Indigenous Nations. Despite the significant outreach, the sampling approach was subject to selection bias, and due to the uniqueness of each Indigenous Nation, the cohort may not be representative of the population of urban Indigenous people, which may limit generalizability. The symptom-based psychiatric rating scales used in this study are validated screening tools which are applied in primary care settings. Screening tools require high sensitivity, as they are designed to provide a rapid indication of whether further assessment is warranted [30]. At a cut-point score of 3, the PHQ-2 and GAD-2 have a high sensitivity of 87% and 86%, respectively [16,17]. However, they are not a substitute for the gold standard for mental health assessment (e.g., in-depth interviews by an experienced mental health professional) and diagnostic tools such as the Diagnostic and Statistical Manual of Mental Disorders (DSM–5). Moreover, there are no known validation studies of the PHQ-2 and GAD-2 in Indigenous Peoples. Finally, the analysis relied on self-reporting, which is subject to recall bias, and causality cannot be established due to the cross-sectional nature of the data, nor can the directionality of results be determined (i.e., people with poor mental health may view their community connections more negatively).

## 5. Conclusions

This study highlights the importance of spirituality and community belonging among urban Indigenous participants, especially within the context of a pandemic. It contributes to our understanding of the important protective role of community belonging for Indigenous Peoples’ mental health and well-being. Improving access to culturally specific programs that foster community belonging may be one avenue to promote mental health, particularly for urban Indigenous Peoples. This would involve improving understanding with regard to Indigenous needs and ways of knowing amongst non-Indigenous service providers, and creating safe spaces within mainstream institutions for individuals to conduct ceremony and receive programming directed to spirit, and funded positions that expand Indigenous-led programs, services and system navigation. In addition, future research (that follows the OCAP^®^ principles) should aim for larger, more representative sample cohorts (including male participants) and more detailed measures of spirituality and spiritual practices (validated by Indigenous Peoples), and should use qualitative approaches such as interviews or Talking Circles (preferably in-person when public health measures allow).

## Figures and Tables

**Table 1 ijerph-19-02472-t001:** Participant demographic characteristics (*n* = 263).

Variable	Frequency (%)
Age	-
16 to 34 Years	100 (38)
35 to 49 Years	94 (36)
50 to 65 Years	56 (21)
65+ Years	13 (5)
Gender	-
Female	231 (88)
Male	23 (9)
Non-binary, Transgender or Two Spirit	9 (3)
Highest Level of Education	-
Primary/Secondary School	83 (32)
College/Other	104 (40)
University	75 (28)
Missing Data	1 (<1)
Housing during COVID	-
Renting	109 (41)
Homeowner	123 (47)
Precarious Housing	31 (12)
Household Annual Income	-
$0 to $39,999	92 (35)
$40,000 to $79,999	85 (32)
≥ $80,000	82 (31)
Missing Data	4 (2)

**Table 2 ijerph-19-02472-t002:** Adjusted logistic regression of the association between the importance of spirituality and sense of community belonging with symptoms of anxiety (GAD-2 score ≥ 3).

Survey Question	Odds Ratio	95% Confidence Interval	*p*-Value
Importance of Spirituality (Pre-Pandemic) Disagree Agree	0.73 Reference	0.32–1.66	0.46
Importance of Spirituality (Peri-Pandemic) Disagree Agree	0.74 Reference	0.37–1.47	0.39
Sense of Community Belonging Weak Strong	2.42 Reference	1.12–5.24	0.02 *

Model adjusted for age, gender, education, and household annual income; * *p* < 0.05.

**Table 3 ijerph-19-02472-t003:** Adjusted logistic regression of the association between the importance of spirituality and sense of community belonging with symptoms of depression (PHQ-2 score ≥ 3).

Survey Question	Odds Ratio	95% Confidence Interval	*p*-Value
Importance of Spirituality (Pre-Pandemic) Disagree Agree	0.46 Reference	0.18–1.12	0.09
Importance of Spirituality (Peri-Pandemic) Disagree Agree	0.99 Reference	0.49–2.01	0.98
Sense of Community Belonging Weak Strong	4.40 Reference	1.95–9.89	0.0004 **

Model adjusted for age, gender, education, and household annual income; ** *p* < 0.001.

**Table 4 ijerph-19-02472-t004:** Frequency of accessing Indigenous cultural services.

Frequency	Count (%)	
	Pre-Pandemic	Post-Pandemic
Never	41 (16)	70 (26)
Infrequently	62 (24)	70 (26)
Frequently	70 (26)	30 (11)
Missing	90 (34)	93 (35)

## Data Availability

The datasets generated and/or analysed during the current study are not publicly available to ensure data integrity and avoid scientific overlap between projects but are available from the corresponding author on reasonable request.

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
