# Peer review of "Spirituality, Community Belonging, and Mental Health Outcomes of Indigenous Peoples during the COVID-19 Pandemic"

_ijerph, 2022, doi:10.3390/ijerph19042472_

Round 1
Reviewer 1 Report
This study presents useful information concerning the relationship between community belonging, spirituality, and self reported symptoms of depression and anxiety among indigenous peoples in Canada during the COVID pandemic. The study employed an online survey to collect data, resulting in a total of 246 completed surveys.
This manuscript has a number of strengths, such as the use of a strength-based approach and the use of previously validated screening tools. In addition, presenting data from an indigenous population is a useful contribution. The content is significant as it presents data from a traditionally under-researched population and displays sensitivity to the issues involved in collecting data from this population. It should be of interest to readers. The methodology is not particularly original (use of online surveys) though it makes a meaningful contribution to the study of the spirituality, community bonding, and mental health among indigenous people in a pandemic.
Its limitations include the use of an online survey instrument, a one item measure of spirituality, the use of a cross sectional approach to data collection, and the inability to take a baseline measure of spirituality, anxiety, or depression.
Spirituality is a complex construct that may have been better measured by more than one item. Since the screening tools have not been validated with indigenous peoples this is also a limitation. This should be noted in the text by the authors.
An additional limitation not mentioned by the authors is self selection of participants, such that those who chose to fill out the survey may have differed in some important respects from those who did not. In addition, generalizability is limited due to the sampling method. Both need to be noted in the text.
The authors should consider further why spirituality did not have any effects. More detailed measures of different dimensions of spirituality may have helped to illuminate possible effects of spirituality or the nonrepresentative sample may have impacted the failure to find effects for spirituality.
I would like to see the authors consider not just implications such as improving access to culturally specific programs that foster community belonging but to briefly identify additional research that should be done. I recognize that this may not be expected by this journal but I find it a weakness of the manuscript in its present form. What else would the authors recommend researchers do to follow up?
For example, given the importance of traditional spiritual practices in terms of their impact on well being and mental health, I would like to see follow up research that includes more nuanced measures of spirituality and spiritual practices and their possible effects on psychological well being. In addition, I would like to see the authors recommend additional research such as using larger and more representative samples as well as qualitative approaches such as interviews. For example, women were over represented. A study that tapped into male respondents may have revealed sex differences.
Reviewer 2 Report
This article explores the association between community belonging and spirituality and mental health outcomes among Indigenous people in Canada during the COVID-19 pandemic. The study is conducted with a cross-sectional observational design. I have the following comments and recommendations to the authors. Very appropriately, the Introduction is quite concise and focused to the issue, but in my opinion some general information about Indigenous People are missing and should be added, for those who are not familiar with the Canadian context and for the sake of clarity. Materials and methods are very adequately described, but it would have been interesting to have more information on the choice of the questions and items, e.g. from a literature reviews, from an experts' panel, etc. The results are clearly presented and well discussed. The topic is niche one but deserves attention.
